# Moderation analysis of subjective well-being, self-efficacy, and academic performance of 4th grade children in Russia

**Diana Akhmedjanova⬥\*, Tatjana Kanonire, Andrey Zakharov⬥**

Department of Educational Programs, Institute of Education, National Research University "Higher School of Economics", Moscow, Russia

\* dakhmedjanova@hse.ru

## Abstract

Research evidence exists on associations of subjective well-being, self-efficacy and various academic and non-academic outcomes with older students; however, there is a research gap on how these variables relate to each other in elementary school students. The goal of this cross-sectional study within a larger longitudinal project was to examine the role of subjective well-being and self-efficacy in predicting academic achievement in math and reading among elementary school students in Russia. The sample included 1,962 students' responses (50.4% girls, 7.7% in rural areas) from elementary schools in central Russia. To measure students' well-being, two subscales from the Survey of Subjective Well-being in School (SSWBS) were used: satisfaction with school ($k=7$) and affect toward school ($k=3$). Domain-specific Self-efficacy Scales: self-efficacy for mathematics ($k=4$) and self-efficacy for reading ($k=4$) were used to measure students' self-efficacy. To measure students' academic achievement in mathematics and reading, the *Progress assessment* – the computer adaptive test – was used. Based on the results, self-efficacy for math relates positively to girls' ($SD=0.32$) and boys' ($SD=0.34$) results in math ($p<0.01$). In contrast, for girls, self-efficacy in reading is more important for their results in reading ($SD=0.27$) than for boys ($SD=0.16$). Hence, both types of self-efficacy are significant for girls and moderate the relationship between subjective well-being and their academic results (satisfaction with school: $SD=0.09$, $p<0.01$; affect toward school: $SD=0.1$, $p<0.05$). The results confirmed the role and importance of self-efficacy in academic performance as well as its relationship to subjective well-being for elementary school students, thus expanding the theoretical views and providing the evidence for the moderating effect of self-efficacy. The paper articulates ideas for future research and implications for teachers and policy makers on targeted development of self-efficacy for math through school wide interventions, especially for elementary school girls.

**Data availability statement:** Data cannot be shared publicly because of the legal regulations accepted within the HSE University. Data are available from the HSE University Data Access (contact Aliya Ermakova, Senior Director of Legal Affairs via email: aermakova@hse.ru) for researchers who meet the criteria for access to confidential data. The authors do not have access to the email provided above. If readers wish to reach out to authors to discuss access to data, they can email the first author of the manuscript at dakhmedjanova@hse.ru.

**Funding:** This study was conducted with support from the Basic Research Program of HSE University as part of the research project "Longitudinal Study of Factors Related to School Failure." The funders had no role in study design, data collection and analysis, decision to publish, or preparation of the manuscript.

**Competing interests:** The authors have declared that no competing interests exist.

## Introduction

Modern education reflects a shift in the educational paradigm from academic achievement alone to holistic personal development and well-being. Research evidence suggests that well-being contributes to various life outcomes such as income, health, and relationships in adulthood [1–3]; however, research on the relationship between subjective well-being in schoolchildren and academic achievement shows a rather weak association [4,5]. In contrast, self-efficacy makes a meaningful contribution to academic performance [6]. Additionally, a growing number of studies suggest positive relationships between well-being and self-efficacy among schoolchildren and university students [7,8]. That is, self-efficacy has a mediating effect between self-concept and well-being [8], positively relates to emotional and cognitive well-being [7], and predicts well-being and academic achievement [9].

Research findings reported above reflect the relationships between self-efficacy and well-being of adolescents and young adults. Additionally, there is generally a lack of studies examining the role of self-efficacy and well-being in elementary schools in the Russian educational context. We could identify only one Russian study, which showed that self-efficacy served as a mediator between reading and subjective well-being of Russian adolescents [10]. There is a clear research gap on how self-efficacy relates to subjective well-being and academic outcomes of elementary students. That is, the novelty and contribution of this study is two-fold. The results of this study will contribute to the theoretical understanding of the moderation effect of self-efficacy, if any, between subjective well-being and academic outcomes of elementary school students. Previous research studies showed that self-efficacy serves as a mediator between various variables [8,11]; however, what role self-efficacy plays or if it strengthens or weakens the effects of well-being on students' academic achievement is less clear. Also, this study will contribute to understanding of the relationships of self-efficacy, academic achievement, and subjective well-being within the elementary school contexts. That is, the results of this study could inform policy makers to adopt school wide policies to train teachers through professional development initiatives on how to incorporate teaching practices supporting the development of self-efficacy and well-being in Russian elementary school contexts.

## Literature review

### Subjective well-being

The amount of research on subjective well-being in the context of education has significantly increased in recent years. In 2015, a well-being survey was added to the Programme for International Student Assessment (PISA) questionnaire, and later, the Organization for Economic Cooperation and Development (OECD) identified well-being as a valuable goal of education [12]. However, research on well-being is complicated by the large number of terms used to describe well-being in different theoretical traditions [4,5,13] as well as by the broad theoretical framework used in international documents. For example, in the OECD framework, well-being is described through five domains: cognitive well-being, psychological well-being (which includes

life satisfaction), physical well-being, social well-being, and material well-being [12]. To understand how well-being is viewed in a particular study, it is often necessary to analyze how the authors have operationalized the construct. In this study, we examine well-being from a hedonistic approach, where well-being is considered as subjective beliefs that an individual's life is pleasant and good. Three components of subjective well-being are recognized: cognitive or life satisfaction, and positive and negative affect [14]. In this study, the special focus is on subjective well-being in school, treated as satisfaction with different school aspects and affect toward school [15].

The study of the relationship between well-being and academic achievement is the most intensively researched field on well-being in children and adolescents [e.g., 16–18]. However, the results of such studies are often contradictory. Some studies show a positive relationship between subjective well-being and academic achievement [e.g., 18,19], while others do not find such a relationship [20] or find differences in well-being only between high and low performers [21]. Inconsistent results could be explained by differences in methodological approaches such as instruments used to measure well-being (e.g., general well-being or well-being in a specific domain; one item or a scale) and academic achievement (GPA, self-evaluation, or standardized test), as well as level of education (elementary or secondary school). The emergence of meta-analyses has made a significant contribution to the debate on the relationships between well-being and academic achievement.

In meta-analysis by Bücker et al. [4], the correlations between academic achievement and subjective well-being were small to medium in magnitude, with overall mean effect $r = 0.16$. Very similar results were obtained in meta-analysis by Kaya and Erdem [5]; although, they included in the analysis not only studies of subjective well-being as life satisfaction in Diener's conceptualization [14], but also within different domains of subjective well-being such as psychological well-being, social well-being, cognitive well-being, and physical well-being. Kaya and Erdem [5] also demonstrate positive but small mean effect size of the relationship between students' well-being and their academic achievement ($r = 0.17$).

The results of meta-analyses suggest that, although not to a strong degree, children's subjective well-being relates to their academic achievement. However, there may be several different mechanisms that explain this relationship. For example, the weak association between well-being and academic achievement may be explained by the inflexibility of the school environment, which links a child's success at school and personal development only to learning outcomes. In this case, such a link between well-being and achievement would have a rather negative connotation. There could also be an alternative positive explanation. Since the main purpose of school is to promote learning, the possibility to fulfill the psychological need of competence through academic achievement could bolster students' well-being.

In the Russian context, there are almost no studies on the relationship between subjective well-being and academic achievement in schoolchildren. The only available study, conducted among primary school students ($n = 144$), found differences in school satisfaction and affect toward school between students with low and high levels of math achievement [21]. Therefore, the present study could provide significant insights into the phenomena of well-being and academic achievement in Russia.

Students' beliefs and perceptions about themselves and their abilities, such as self-efficacy, could represent another source of explanation. Therefore, to better understand the mechanism of the relationship between well-being and academic outcomes, this study will test the moderating effect of self-efficacy on the relationship between well-being and academic achievement.

## Self-efficacy

Self-efficacy is another source of motivation that relates to competence. Students' self-efficacy is their perceived confidence to do a task and study well [22–24]. Research studies show that self-efficacy positively relates to students' academic outcomes and psychological constructs of motivation, self-regulated learning (SRL), and subjective well-being [6,8,23,25–27]. Self-efficacy changes depending on tasks, mastery experiences, and successes or failures [25]. In addition, self-efficacy and academic achievement show a reciprocal relationship [6]. That is, high academic achievement

contributes to an increase in self-efficacy, and high self-efficacy relates to higher academic achievement, as is evidenced by the results of the meta-analysis of studies ($n = 11$) examining self-efficacy and academic outcomes [6].

High self-efficacy tends to be a strong predictor of students' achievement and success [6,27]. Generally, students with high self-efficacy exhibit higher motivation and academic achievement, seek more opportunities to learn, regulate their own learning, interpret their academic failures as a lack of effort, and perceive learning difficulties as challenges to resolve than students with low self-efficacy [6,23–25,27].

Researchers debate over how to conceptualize and measure self-efficacy, stating that self-efficacy can be domain-specific or domain-general [27]. Bandura [23] outlined three levels of specificity when discussing self-efficacy. The general level refers to students' beliefs about their general academic self-efficacy to study or general self-efficacy within a domain, such as "I am sure I can do well in math." The intermediate level refers to beliefs about certain skills or competencies, such as "I am sure I can solve math equations." Finally, the specific level of self-efficacy focuses on a task, such as "I am sure I can solve this quadratic equation." Irrespective of the views, Bandura [23] emphasized the importance of assessing "… the multifaceted ways in which efficacy beliefs operate within the selected activity domain" (p. 310). Talsma and colleagues [6] identified in their meta-analysis that the effect sizes for scales measuring domain-specific self-efficacy were higher than for scales measuring general self-efficacy.

In the Russian context, a few studies examined self-efficacy among university students [28–30] and focused on foreign students learning Russian [31]. For instance, the study of the third-year university students ($n = 558$) indicated medium levels of self-efficacy; however, the group with the medium level of self-efficacy showed higher levels of job stability than students with low levels of self-efficacy [29]. Several studies examined Russian middle school students' self-efficacy and academic achievement [32,33], including relationships with SRL [34] and science identity [35]. Belova and colleagues [35] examined the relationship of science learning self-efficacy with science identity, checking for the moderation effects of student gender and science success among Russian high school students ($n = 519$). The results of the study supported the significant link between science communication efficacy and science identity. Also, gender did not show a moderating effect between self-efficacy and science identity. Finally, science success did not significantly moderate this link; however, interaction of science success with science communication efficacy resulted in the negative effect on science identity of high school students. Belova and colleagues [35] speculate that a nuanced moderation effect of science success may suggest that students who perform well in science but do not feel confident in communicating about science may experience negative effects on developing their science identity.

A brief overview of studies in the Russian context show that self-efficacy is researched among middle or high school students, and mostly cover the samples of university students. Hence, similar to the foreign research studies, Russian findings focus mostly on relationships of self-efficacy with the academic results, job satisfaction, coping strategies, but not really on well-being and samples of elementary school students.

## Subjective well-being and self-efficacy

Research studies in the field of education provide compelling evidence of the effects and correlations of self-efficacy with academic outcomes [25,27,34,35]. In recent years, researchers have also shifted their attention to other psychological constructs, such as subjective well-being.

Research studies show that self-efficacy has a mediating effect between students' self-concept and subjective well-being on a sample of migrant and native Chilean adolescents [$n = 406$; 8]. A study conducted among Italian high school and university students ($n = 465$) during the COVID-19 lockdown identified that self-efficacy for self-regulated learning and self-efficacy for positive and negative emotions were related to cognitive and emotional well-being [7]. Yet another study of German 8th and 9th grade students ($n = 767$) identified that positive school climate, self-efficacy, and test anxiety predicted subjective well-being and academic achievement [9].

In the Russian context, we could identify a comparative study examining well-being and self-efficacy in Russian and Kazakh students [36] and a master's degree thesis that examined the well-being and self-efficacy of Russian adolescents [10]. The master's thesis examined direct and indirect effects between self-efficacy in reading and subjective well-being on a sample of more than 15,000 Russian adolescents from the PISA-2018 assessment and showed that general self-efficacy served as a mediator between self-efficacy in reading and subjective well-being [10]. In addition, reading self-efficacy had a positive effect on students' subjective well-being.

Similar to the studies conducted around the world, there was an attempt to examine the mediating effect of self-efficacy between SRL and academic results in a sample of Russian elementary students [11]. Akhmedjanova [11] examined the relationships among SRL, self-efficacy, and academic outcomes (math and reading) of the fourth-grade students ($n = 2,158$) using structural equation modelling [11]. While the results indicated the negative direct effects of SRL on academic outcomes, self-efficacy served as a mediator between SRL and academic achievement in both math and reading models in the analyses by gender and location. In this way, self-efficacy serves as a mediator between SRL and academic outcomes of elementary school students in Russia. However, it is not clear how it relates to subjective well-being.

## Current study

Studies in other countries provide evidence of relationships between self-efficacy and subjective well-being, stating specifically that it serves as a mediator between the psychological constructs and academic outcomes. In addition, most of these studies both in Russia and across the world are conducted with samples of adolescents and young adults. While there are many studies examining self-efficacy and various academic and non-academic outcomes with older students, there is a research gap on how these variables relate to each other in elementary school students. Therefore, the goal of this paper was to examine the role of subjective well-being in school and self-efficacy in predicting academic achievement in math and reading among elementary school students in Russia (Fig 1).

To achieve the research goal, we addressed the following research questions:

1. What is the relationship between students' outcomes in math and reading and their subjective well-being in school and domain-specific self-efficacy?

2. Is there a moderation effect of domain-specific self-efficacy on the relationship between students' results in math and reading and their subjective well-being in school?

3. Are there gender differences in the relationship between the student's test scores in math and reading and their subjective well-being in school and domain-specific self-efficacy?

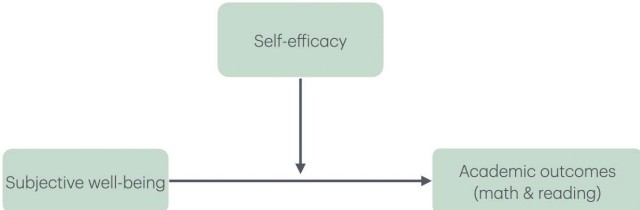

**Fig 1. The Moderating Effect of Domain-specific Self-efficacy between Subjective Well-being in School and Academic Outcomes of Elementary School Students.**

## Methods

### Study design

This study is part of the longitudinal project examining the factors contributing to students' academic failure in Russian primary schools. For the purposes of this study, we used survey data from elementary students in the first wave of data collection in fall 2022.

### Procedures

The data collection took place online in 40 public schools in one of the regions of central Russia after receiving approval from the Ethics Committee (#19). Parents were informed about the purpose of the study and signed online consent forms. Before starting the assessment, children provided their assent to participate in the study. Data were collected on two separate days: one day for math and reading assessments and another day for survey data.

### Sample

To recruit participants, we used the random stratified proportional sampling at the level of schools. That is, the unit of selection was an educational organization – school. For school to be included into the sample, it should have had a within school support system to address school failure across urban and rural areas, and 42 schools fit this criterion. As a result, nine municipalities in a central region of Russia were included in the study across three types of locations: a large city, small towns, and rural areas, resulting in 40 schools that agreed to participate and were able to provide research data. The sample included responses from the fourth graders, 9 and 10 years old. The missing data analysis revealed that the dataset included 13% or less of missing cases in variables underpinning constructs of self-efficacy in math and reading, affect towards school, and satisfaction with school. The missing values originated due to students skipping specific questions in these constructs. Rigorous approaches such as multiple imputation require appropriate auxiliary data that are predictive of both the missingness and the missing values to produce reliable imputations [37]. The dataset provides only limited auxiliary information – students' gender, outcome variables (test scores), and location. Given the limited auxiliary variables, we opted for listwise deletion to preserve the validity and interpretability of the findings while acknowledging its limitations. The missing data were removed using listwise deletion, resulting in the 1,962 students' responses (50.4% girls, 7.7% in rural areas), which were used for the analysis. The listwise deletion was chosen since the sensitivity analyses indicated that the main results were robust to this choice.

**Instruments.** To measure students' well-being, two subscales from the Survey of subjective well-being in school (SSWBS) were used: satisfaction with school (k = 7) and affect toward school (k = 3) [15]. The Likert-type agreement scale ranging from 1 to 4 was used for satisfaction with the school scale, and the Likert-type frequency scale ranging from 1 to 4 was used for affect toward the school scale. The reliability indices revealed appropriate estimates ($\alpha = 0.87$; $\alpha = 0.89$).

The domain-specific self-efficacy scales were developed for the longitudinal project [38]. In this study, we used two scales: self-efficacy for mathematics (k = 4) and self-efficacy for reading (k = 4) on a four-point Likert-type scale (4 – I can do it well, 1– I cannot do it at all). An example item: *"Can you solve a math problem?".* The internal consistency for both scales was good: $\alpha_{math} = 0.8$; $\omega_{math} = 0.81$; and $\alpha_{read} = 0.78$; $\omega_{read} = 0.8$.

To measure students' academic achievement in mathematics and reading, the *Progress assessment* was used. *Progress* is the computer adaptive test to measure students' mathematics and reading achievement [39,40]. The mathematics domain measures spatial concepts, measurement, patterns and sequences, modeling, and data handling and includes 30 dichotomous multiple-choice items with acceptable psychometric properties [40]. The internal consistency of the whole scale indicated high reliability estimates, $\alpha = 0.88$; $\omega = 0.89$. The reading test measures students' skills of searching for information, making interpretations, analyzing, and synthesizing, and includes 3 texts with 17 items [39]. The internal consistency of the whole scale indicated acceptable reliability estimates, $\alpha = 0.63$; $\omega = 0.67$. Generally, the reliability estimates

of > 0.70 are acceptable for conducting further analyses. However, some scholars establish acceptable reliability estimates at 0.65 [41] or at 0.6 [42]. Due to the reading test reliability estimates falling between 0.63 and 0.67, we concluded that the estimates were acceptable to conduct further analyses; however, the results of the reading model should be interpreted with caution.

Data also provides some students' characteristics. In our analysis we controlled for students' gender (1 = girl) and school location in rural area (1 = yes). In the models with school fixed effect we used schools' ids. Descriptive statistics are presented in Table 1.

## Data analysis

To conduct the analysis, the authors accessed the dataset on 10/04/2024. The analysis was done in three consecutive steps. First, to estimate the relationship between students' outcomes in math and reading, their self-efficacy in these subjects, and their subjective well-being, we applied regular linear regression (ordinary least squares, or OLS):

$$Test_{is} = \alpha + \beta WB_{is} + \gamma SE_{is} + \delta Cov_{is} + \theta Sch_s + e_{is}, \tag{1}$$

where $i$ denotes a student, $s$ denotes a school. $Test_{is}$ represents a student's test score in math or reading. $WB_{is}$ is a student's well-being score (satisfaction with school or affect toward school). $SE_{is}$ is student's self-efficacy in a subject (math or reading). $Cov_{is}$ represents a vector of covariates including students' gender and school location in rural area. $Sch_s$ is a school fixed effect. $\alpha$, $\beta$, $\gamma$, $\delta$, $\theta$ is a vector of regression coefficients. $e_{is}$ is an error term.

The intra-class correlation coefficient (ICC), estimated from random-intercept null models, was 0.375 for math scores and 0.115 for reading scores, indicating substantial and moderate between-school variance, respectively. Although this suggests clustering at the school level, particularly for math scores, the primary analyses focused on models with school fixed effects to facilitate interpretation of moderation effects at the individual level. The use of school fixed effects is a common and rigorous approach in education research, as it controls for observed and unobserved differences between schools.

Second, to estimate the moderation effect of students' self-efficacy, we added an interaction between variables of self-efficacy and well-being to the right side of the equation (1):

$$Test_{is} = \alpha' + \beta' WB_{is} + \gamma' SE_{is} + \sigma' WB_{is} * SE_{is} + \delta' Cov_{is} + \theta' Sch_s + e'_{is}, \tag{2}$$

Third, to test gender differences in the relationship between the student's math and reading scores and their subjective well-being and self-efficacy, we ran the analysis (equations 1 and 2) on subsamples of girls and boys.

**Table 1. Descriptive Statistics (n = 1962).**

| Variable | Mean | SD |
|---|---|---|
| Math | −0.221 | 1.128 |
| Reading | −0.976 | 0.647 |
| Self-Efficacy in Math | 0.000 | 0.913 |
| Self-Efficacy in Reading | 0.000 | 0.914 |
| Satisfaction with School | 0.000 | 0.915 |
| Affect to School | 0.000 | 0.909 |
| Girl (y/n) | 0.504 | 0.500 |
| Rural area (y/n) | 0.077 | 0.267 |

At all stages, the analysis was done separately for each well-being variable (satisfaction with school or affect toward school) due to a high correlation between them in this study (r = 0.97). For the same reason, we did not use both self-efficacy in math and reading (r = 0.82) in the same model. We opted not to include both well-being subscales or both self-efficacy domains simultaneously in the same models to avoid issues of multicollinearity. By modeling these constructs separately, we aimed to provide clearer, more reliable estimates of their distinct associations and potential moderation effects. Other predictors were added stepwise. This strategy allowed for determining the changes in the coefficients of the main predictors.

In the tested models, $\beta$, $\gamma$, $\beta'$, $\gamma'$, and $\delta'$ coefficients could be regarded as unbiased estimates of causal effect, based on a strict assumption that well-being and self-efficacy variables are not related to the error terms. To decrease bias, we controlled for available students' characteristics (gender and rural area) because they could relate to both students' outcomes and the main predictors. We also fixed the school effect to account for all differences between schools. However, there could still be extraneous variables related to the outcomes and predictors, which is a limitation of the analysis.

The data analysis was conducted in R using the lm function.

## Results

1. What is the relationship between students' outcomes in math and reading and their subjective well-being and domain-specific self-efficacy?

The estimates of the relationship of test scores in math and reading to students' subjective well-being and self-efficacy are presented in Tables 2 and 3 (Columns 1–8). For both types of well-being, the same tendency is observed across subjects. With no control for other variables (Tables 2 and 3, Columns 1 & 5), both types of well-being show a statistically significant ($p < 0.01$) but weak relationship (0.13–0.15 SD) with students' outcomes. For affect toward school, this relationship is a little stronger for both subjects. The relationship with math is a little stronger compared

**Table 2. Relationship of Math and Readings Scores with Students' Affect to School and Self-Efficacy, Whole Sample (OLS Estimates).**

| Variable | Math | | | | Reading | | | |
|---|---|---|---|---|---|---|---|---|
| | (1) | (2) | (3) | (4) | (5) | (6) | (7) | (8) |
| Intercept | 0.000 | 0.000 | −0.013 | 1.508*** | 0.000 | 0.000 | −0.002 | 0.706*** |
| | (0.022) | (0.021) | (0.022) | (0.153) | (0.022) | (0.022) | (0.023) | (0.186) |
| Affect to school | 0.150*** | 0.038 | 0.041 | −0.033 | 0.131*** | 0.053* | 0.054* | 0.011 |
| | (0.025) | (0.024) | (0.024) | (0.021) | (0.025) | (0.026) | (0.026) | (0.025) |
| Self-efficacy | | 0.386*** | 0.391*** | 0.333*** | | 0.237*** | 0.238*** | 0.216*** |
| | | (0.024) | (0.024) | (0.021) | | (0.025) | (0.026) | (0.025) |
| Girl (y/n) | | | | −0.029 | | | | 0.123** |
| | | | | (0.035) | | | | (0.043) |
| Rural (y/n) | | | | 0.139 | | | | 0.261 |
| | | | | (0.410) | | | | (0.498) |
| Affect to school x Self-efficacy | | | 0.055* | 0.028 | | | 0.007 | −0.014 |
| | | | (0.023) | (0.019) | | | (0.024) | (0.023) |
| School Fixed Effect | No | No | No | Yes | No | No | No | Yes |
| n | 1962 | 1962 | 1962 | 1962 | 1962 | 1962 | 1962 | 1962 |
| $R^2$ | 0.019 | 0.132 | 0.135 | 0.440 | 0.014 | 0.056 | 0.056 | 0.174 |
| $\Delta R^2$ | | 0.113 | 0.003 | 0.305 | | 0.042 | 0.000 | 0.118 |

*Notes:* \* $p < 0.05$, \*\* $p < 0.01$, \*\*\* $p < 0.001$.

**Table 3. Relationship of Math and Reading Scores with Students' Satisfaction with School and Self-Efficacy, Whole Sample (OLS Estimates).**

| Variable | Math | | | | Reading | | | |
|---|---|---|---|---|---|---|---|---|
| | (1) | (2) | (3) | (4) | (5) | (6) | (7) | (8) |
| Intercept | 0.000 | 0.000 | −0.011 | 1.507*** | 0.000 | 0.000 | 0.000 | 0.705*** |
| | (0.022) | (0.021) | (0.022) | (0.154) | (0.022) | (0.022) | (0.023) | (0.186) |
| Satisfaction with school | 0.132*** | 0.023 | 0.029 | −0.031 | 0.119*** | 0.044 | 0.044 | 0.012 |
| | (0.025) | (0.024) | (0.024) | (0.020) | (0.025) | (0.025) | (0.025) | (0.025) |
| Self-efficacy | | 0.390*** | 0.394*** | 0.332*** | | 0.241*** | 0.241*** | 0.215*** |
| | | (0.024) | (0.024) | (0.021) | | (0.025) | (0.025) | (0.025) |
| Girl (y/n) | | | | −0.029 | | | | 0.123** |
| | | | | (0.035) | | | | (0.043) |
| Rural (y/n) | | | | 0.143 | | | | 0.266 |
| | | | | (0.410) | | | | (0.497) |
| Satisfaction with school x Self-efficacy | | | 0.047* | 0.023 | | | 0.001 | −0.018 |
| | | | (0.022) | (0.018) | | | (0.023) | (0.022) |
| School Fixed Effect | No | No | No | Yes | No | No | No | Yes |
| $n$ | 1962 | 1962 | 1962 | 1962 | 1962 | 1962 | 1962 | 1962 |
| $R^2$ | 0.014 | 0.132 | 0.134 | 0.440 | 0.012 | 0.056 | 0.056 | 0.174 |
| $\Delta R^2$ | | 0.118 | 0.002 | 0.306 | | 0.044 | 0.000 | 0.118 |

Notes: * $p < 0.05$, ** $p < 0.01$, *** $p < 0.001$.

to reading for both types of well-being. When controlling for students' self-efficacy, the relationship between both types of well-being with math (Tables 2 and 3, Column 2) and the relationship between satisfaction with school and reading (Table 3, Column 6) becomes statistically not different from zero. The relationship between reading scores and affect toward school (Table 2, Column 6) also declines but remains statistically significant (0.05 SD, $p < 0.05$) until we add control for other variables (Table 2, Column 8). Finally, in all the models with school fixed effect (Tables 2 and 3, Column 8), the relationship between both types of well-being and both types of students' outcomes is not statistically significant.

As for self-efficacy, it shows a robust and statistically significant relationship ($p < 0.01$) with students' outcomes in both math and reading, indicating a stronger relationship with math. When controlling for demographic characteristics and adding the school fixed effect, we detect a small decrease in estimates. The coefficient declines from 0.39 to 0.33 SD and from 0.24 to 0.22 SD for math and reading, respectively (Tables 2 and 3, Columns 2–8).

2. Is there a moderation effect of domain-specific self-efficacy on the relationship between students' results in math and reading and their subjective well-being?

To address this research question, we added an interaction of domain-specific self-efficacy with both types of well-being to the models. With no control for demographic characteristics (Tables 2 and 3, Column 3), the results indicate a small statistically significant interaction effect with both affect toward school (0.06 SD, $p < 0.05$, $\Delta R^2 = 0.003$, S1) and satisfaction with school (0.05 SD, $p < 0.05$, $\Delta R^2 = 0.003$, S3), but only for math. The interaction effect was not statistically significant with affect toward school (S2) and satisfaction with school (S4) for reading (Tables 2 and 3, Column 7). When adding covariates, the interaction effect becomes insignificant across the models (Tables 2 and 3, Column 7) yet indicating the moderate effect sizes for math ($\Delta R^2 = 0.305$) and small effect size for reading ($\Delta R^2 = 0.118$, Table 2, Columns 4 & 8) and with the moderate effect sizes for math ($\Delta R^2 = 0.306$) and small effect size for reading ($\Delta R^2 = 0.118$, Table 3, Columns 4 & 8).

 

3. Are there gender differences in the relationship between the student's test scores in math and reading and their subjective well-being and domain-specific self-efficacy?

Comparisons across genders show a similar relationship between test scores and well-being. That is, both types of well-being significantly ($p < 0.01$) relate to students' outcomes in math and reading for girls and boys, only with no control for other variables (Tables 4–7, Columns 1& 5). However, for girls, the effect sizes of both types of well-being are a little higher for math (0.16 and 0.17 SD for satisfaction with school and affect toward school, respectively) compared to reading (0.1 and 0.12 SD; Tables 4, 6, Columns 1, 5). For boys, this is true only for affect toward school, although the estimates are quite comparable (0.14 and 0.13 SD for math and reading; Table 7, Columns 1, 5); for satisfaction with school, the difference in the estimates across subjects is neglectable (Table 5, Columns 1, 5).

The relationship between self-efficacy and students' outcomes is positive and statistically significant ($p < 0.001$) across models (Tables 4–7, Columns 2–4, 6–8). The effect sizes for math are higher than for reading for both boys and girls. However, for boys, when controlling for other variables and adding the school fixed effect, the effect size for math (0.34 SD) is almost twice as high as compared to reading (0.16 SD; Tables 5, 7, Columns 4, 8). For girls, the difference between self-efficacy effects for math (0.32–0.33 SD) and reading (0.27–0.28 SD) is not that large (Tables 4, 6, Columns 4, 8).

Compared to the results on the whole sample, the interaction effects on subsamples are more nuanced. For girls, there is a statistically significant interaction effect for math only (Tables 4, 6, Columns 3–4, 7–8). The interaction effect of satisfaction with school and self-efficacy is small (0.09 SD) and significant ($p < 0.01$) only when covariates are not added (Table 4, Columns 3–4), yet moderate effect size for the school fixed effect ($\Delta R^2 = 0.339$). In contrast, the interaction effect of affect toward school and self-efficacy is statistically significant ($p < 0.05$) both without and with control for other variables and the school fixed effect (0.1 and 0.06 SD, respectively; Table 6, Columns 3–4), again indicating a moderate effect size for the school fixed effect ($\Delta R^2 = 0.337$). The interaction effects for boys are not significantly different from zero for both math and reading (Tables 5, 7, Columns 3–4, 7–8).

## Discussion and conclusion

The primary goal of this study was to examine the role of subjective well-being in school and self-efficacy in predicting academic achievement in math and reading among elementary school students in Russia. The results indicated that both

**Table 4. Relationship of Test Scores with Students' Satisfaction with School and Self-Efficacy, Girls (OLS Estimates).**

| Variable | Math | | | | Reading | | | |
|---|---|---|---|---|---|---|---|---|
| | (1) | (2) | (3) | (4) | (5) | (6) | (7) | (8) |
| Intercept | −0.012 | 0.043 | 0.019 | 1.616*** | −0.008 | 0.019 | 0.025 | 0.953*** |
| | (0.032) | (0.030) | (0.032) | (0.217) | (0.032) | (0.031) | (0.033) | (0.264) |
| Satisfaction with school | 0.157*** | 0.020 | 0.043 | −0.025 | 0.102** | −0.018 | −0.022 | −0.057 |
| | (0.037) | (0.037) | (0.038) | (0.031) | (0.037) | (0.038) | (0.039) | (0.038) |
| Self-efficacy | | 0.393*** | 0.393*** | 0.323*** | | 0.305*** | 0.304*** | 0.277*** |
| | | (0.035) | (0.035) | (0.030) | | (0.036) | (0.036) | (0.035) |
| Rural (y/n) | | | | −0.594 | | | | −0.334 |
| | | | | (0.571) | | | | (0.695) |
| Satisfaction with school x Self-efficacy | | | 0.090** | 0.051 | | | −0.018 | −0.046 |
| | | | (0.034) | (0.028) | | | (0.034) | (0.033) |
| School Fixed Effect | No | No | No | Yes | No | No | No | Yes |
| $n$ | 988 | 988 | 988 | 988 | 988 | 988 | 988 | 988 |
| $R^2$ | 0.018 | 0.129 | 0.135 | 0.474 | 0.008 | 0.076 | 0.076 | 0.221 |
| $\Delta R^2$ | | 0.111 | 0.006 | 0.339 | | 0.068 | 0.000 | 0.145 |

*Notes: * $p < 0.05$, ** $p < 0.01$, *** $p < 0.001$.*

**Table 5. Relationship of Test Scores with Students' Satisfaction with School and Self-Efficacy, Boys (OLS Estimates).**

| Variable | Math | | | | Reading | | | |
|---|---|---|---|---|---|---|---|---|
| | (1) | (2) | (3) | (4) | (5) | (6) | (7) | (8) |
| Intercept | 0.010 | −0.042 | −0.045 | 1.374*** | 0.010 | −0.005 | −0.007 | 0.624* |
| | (0.032) | (0.030) | (0.031) | (0.215) | (0.032) | (0.032) | (0.033) | (0.260) |
| Satisfaction with school | 0.123*** | 0.031 | 0.031 | −0.030 | 0.128*** | 0.077* | 0.078* | 0.053 |
| | (0.033) | (0.032) | (0.032) | (0.028) | (0.033) | (0.034) | (0.034) | (0.034) |
| Self-efficacy | | 0.384*** | 0.386*** | 0.337*** | | 0.195*** | 0.196*** | 0.160*** |
| | | (0.034) | (0.034) | (0.030) | | (0.036) | (0.036) | (0.036) |
| Rural (y/n) | | | | 0.909 | | | | 0.867 |
| | | | | (0.590) | | | | (0.710) |
| Satisfaction with school x Self-efficacy | | | 0.014 | −0.006 | | | 0.006 | −0.011 |
| | | | (0.030) | (0.026) | | | (0.032) | (0.031) |
| School Fixed Effect | No | No | No | Yes | No | No | No | Yes |
| $n$ | 974 | 974 | 974 | 974 | 974 | 974 | 974 | 974 |
| $R^2$ | 0.014 | 0.129 | 0.129 | 0.435 | 0.015 | 0.044 | 0.044 | 0.174 |
| $\Delta R^2$ | | 0.115 | 0.000 | 0.306 | | 0.029 | 0.000 | 0.130 |

Notes: * $p<0.05$, ** $p<0.01$, *** $p<0.001$.

**Table 6. Relationship of Test Scores with Students' Affect to School and Self-Efficacy, Girls (OLS Estimates).**

| Variable | Math | | | | Reading | | | |
|---|---|---|---|---|---|---|---|---|
| | (1) | (2) | (3) | (4) | (5) | (6) | (7) | (8) |
| Intercept | −0.014 | 0.042 | 0.015 | 1.619*** | −0.010 | 0.017 | 0.020 | 0.940*** |
| | (0.032) | (0.030) | (0.032) | (0.217) | (0.032) | (0.031) | (0.033) | (0.264) |
| Affect to school | 0.172*** | 0.031 | 0.049 | −0.029 | 0.124*** | 0.003 | 0.002 | −0.040 |
| | (0.036) | (0.037) | (0.037) | (0.031) | (0.037) | (0.038) | (0.039) | (0.038) |
| Self-efficacy | | 0.389*** | 0.390*** | 0.325*** | | 0.297*** | 0.297*** | 0.273*** |
| | | (0.035) | (0.035) | (0.030) | | (0.036) | (0.036) | (0.035) |
| Rural (y/n) | | | | −0.594 | | | | −0.342 |
| | | | | (0.570) | | | | (0.695) |
| Affect to school x Self-efficacy | | | 0.097** | 0.057* | | | −0.008 | −0.038 |
| | | | (0.034) | (0.028) | | | (0.034) | (0.033) |
| School Fixed Effect | No | No | No | Yes | No | No | No | Yes |
| $n$ | 988 | 988 | 988 | 988 | 988 | 988 | 988 | 988 |
| $R^2$ | 0.022 | 0.129 | 0.137 | 0.474 | 0.012 | 0.075 | 0.075 | 0.220 |
| $\Delta R^2$ | | 0.107 | 0.008 | 0.337 | | 0.063 | 0.000 | 0.145 |

Notes: * $p<0.05$, ** $p<0.01$, *** $p<0.001$.

satisfaction with school and affect toward school were significantly and positively related to students' outcomes in math and reading, without controlling for other variables. However, this relationship is weak, which corresponds with the results of previous studies [4,5].

The models that examined both types of subjective well-being did not show any relation to students' results in math when controlled for self-efficacy. That is, self-efficacy in math fully explained the relationship of satisfaction with school and affect toward school with students' results in math. As for reading, the two types of subjective well-being showed

**Table 7. Relationship of Test Scores with Students' Affect to School and Self-Efficacy, Boys (OLS Estimates).**

| Variable | Math | | | | Reading | | | |
|---|---|---|---|---|---|---|---|---|
| | (1) | (2) | (3) | (4) | (5) | (6) | (7) | (8) |
| Intercept | 0.012 | −0.040 | −0.044 | 1.373*** | 0.010 | −0.005 | −0.008 | 0.634* |
| | (0.032) | (0.030) | (0.031) | (0.215) | (0.032) | (0.032) | (0.033) | (0.260) |
| Affect to school | 0.144*** | 0.048 | 0.047 | −0.030 | 0.131*** | 0.078* | 0.078* | 0.041 |
| | (0.034) | (0.033) | (0.033) | (0.028) | (0.034) | (0.035) | (0.035) | (0.034) |
| Self-efficacy | | 0.379*** | 0.382*** | 0.337*** | | 0.195*** | 0.196*** | 0.164*** |
| | | (0.034) | (0.034) | (0.030) | | (0.036) | (0.037) | (0.036) |
| Rural (y/n) | | | | 0.912 | | | | 0.862 |
| | | | | (0.590) | | | | (0.711) |
| Affect to school x Self-efficacy | | | 0.019 | −0.005 | | | 0.011 | −0.011 |
| | | | (0.032) | (0.027) | | | (0.033) | (0.033) |
| School Fixed Effect | No | No | No | Yes | No | No | No | Yes |
| n | 974 | 974 | 974 | 974 | 974 | 974 | 974 | 974 |
| $R^2$ | 0.019 | 0.13 | 0.13 | 0.435 | 0.015 | 0.044 | 0.044 | 0.173 |
| $\Delta R^2$ | | 0.111 | 0.000 | 0.305 | | 0.029 | 0.000 | 0.129 |

Notes: * $p < 0.05$, ** $p < 0.01$, *** $p < 0.001$.

different results. When controlling for self-efficacy, the analysis revealed that the relationship between affect toward school and students' results in reading was statistically significant, yet weak. However, when the model accounted for students' gender, this relationship became insignificant. In contrast, the relationship between satisfaction with school and reading became non-significant when controlling for self-efficacy, even without control for other students' characteristics.

A more detailed analysis revealed differences based on students' gender. The relationship between both types of subjective well-being and students' results in math and reading was statistically significant for both genders, not controlling for self-efficacy. However, the moderation effect of self-efficacy could be observed for both domains of well-being and students' outcomes only in math and only for girls. When controlling for students' characteristics, the described moderation effect remained significant only for the scale of affect toward school.

The results of this study are partially in line with the previous studies, showing gender differences with boys scoring higher in mathematics, including standardized assessments such as PISA [43] and research on perceived self-efficacy [44]. In the current study, both self-efficacy for math and reading are lower for girls than for boys. Interestingly, in the sample of Russian elementary students, self-efficacy for reading is higher for boys than girls, which differs from the results of other studies [26]. However, Peura and colleagues [45] also reported on a study with boys having higher levels of reading self-efficacy, which might have been due to measuring recreational rather than school-based reading self-efficacy. Pajares and Valiante [44] argued that gender differences in academic motivation, including self-efficacy, typically favor girls but might be the result of stereotypical gender roles and beliefs, which might be deeply rooted in how teachers, family, and society try to emphasize the importance of math for boys rather than girls [45]. To support this claim, Peura et al. [45] emphasize that differences in reading self-efficacy between girls and boys are typically small and encourage measuring individual rather than group differences in self-efficacy across gender. In this way, the results of our study may indicate that potential classroom practices; specifically, how teachers provide feedback on self-efficacy might be favoring boys rather than girls in mathematics and reading in this sample. That is, girls in Russian classrooms might typically follow teacher's instructions more eagerly, behave and study better than boys, including in elementary school settings. As a result, teachers might not pay as much attention to girls and provide as detailed feedback to them as they do for boys. Similar behaviors could also be reflected in parental expectations of girls being docile and well-behaved and boys being

rowdy and unruly, which might further cement specific gender roles and expectations among Russian elementary school students. However, these assumptions need close examination in classroom and family settings. It is also not clear how teachers and school contexts develop and support subjective well-being in the researched schools, which should become areas of future research, for instance, on teacher's perceptions of gender roles.

A possible explanation of the lack or low moderation effects of self-efficacy on the relationship between subjective well-being and students' academic results could be that students in the fourth grade are still too young to objectively report on their confidence in correctly solving a math problem or understanding new words from text. Neuropsychological studies provide compelling evidence that abstract thinking and analyzing skills necessary to reflect on students' behaviors and emotions are still developing at this age [46]. Also, the data were collected at the beginning of the academic year, which might suggest that students' characteristics and academic achievement were more likely to represent their development, knowledge, and skills in their third grade than their fourth grade.

## Limitations

While this study explained the nature of the relationship between subjective well-being, self-efficacy, and academic performance among Russian schoolchildren, it has some inherent limitations. The study was cross-sectional and included only the results of the fourth-grade students in one of the regions of Russia. In addition, self-report questionnaires were used to measure subjective well-being and self-efficacy, which typically identify average results across participants and have low to medium correlations with academic outcomes [23]. Also, the reliability estimates of the reading assessment were below the established > 0.70 estimate; therefore, the results of the reading models should be interpreted with caution. For the analysis, we used only children's outcomes, which might limit the role of parental and teacher characteristics, socio-economic status, school context, and other important variables. We did not measure or inquire about how teachers promoted and developed students' subjective well-being and self-efficacy in various school contexts. Finally, some research evidence suggests that high academic performance leads to increases in domain-specific self-efficacy [25], suggesting that there can be bidirectional effects and calling for further research.

## Future research

Therefore, as future research, we recommend examining the development of students' well-being and self-efficacy as well as their interplay in relation to academic results longitudinally, using more psychometrically robust instruments to measure reading skills and checking for their bidirectional associations. The substantial between-school variance observed for math scores suggests that future research could further examine these relationships using multilevel models. It is also worth examining the school context and educational programs to identify how they promote the development of well-being and self-efficacy. Along these lines, it will be beneficial to conduct intervention studies with students of various ages to target the development of well-being and self-efficacy. Also, adding school, teaching, and parental characteristics to children's outcomes might provide a more comprehensive understanding of the interplay between well-being, self-efficacy, and academic outcomes. We also call for qualitative studies which might potentially shed light on possible gender roles in school contexts, focusing on parental, teachers', and children's perceptions, behaviors, and practices in and outside of class in relation to self-efficacy and subjective well-being.

## Implications for practice

Irrespective of the limitations outlined above and small effect sizes, we confirmed the role and importance of self-efficacy in academic results as well as its relationship to subjective well-being for elementary school students in Russia. Based on the results, self-efficacy for math is equally important for girls' and boys' results in math. In contrast, for girls, self-efficacy in reading is more important for their results in reading than for boys. Hence, both types of self-efficacy are important for girls and moderate the relationship between subjective well-being and their academic results. Therefore, one of the

 

practical implications is to target the development and increase in self-efficacy for math, especially for elementary school girls. That is, it is important to create opportunities of professional development for teachers so they can learn about how to increase and sustain high levels of self-efficacy through such practices as creating situations of academic success by assigning tasks within students' zone of proximal development, providing supportive feedback on how to do tasks, and acknowledging students' emotional reactions to successes and failures, to name a few [23].

Hence, teachers should be trained on how to deliver constructive feedback regarding math to girls, since it might potentially lead to an improvement in their self-efficacy for math and, further, increase their subjective well-being and math performance. It is especially important since research findings suggest a decreasing trend in subjective well-being in adolescence [47,48]. It is also worth improving self-efficacy for reading since it is one of the basic literacy skills that should be mastered to tackle tasks across subjects in school. To support the proposed practical implications, it is worth drawing the attention of policy makers to assist with the development and application of professional development opportunities for elementary school teachers.

## Supporting information

**S1 Fig. The Moderation Effects of Self-Efficacy between Satisfaction with School and Math Scores.**
(TIFF)

**S2 Fig. The Moderation Effects of Self-Efficacy between Satisfaction with School and Reading Scores.**
(TIFF)

**S3 Fig. The Moderation Effects of Self-Efficacy between Affect to School and Math Scores.**
(TIFF)

**S4 Fig. The Moderation Effects of Self-Efficacy between Affect to School and Reading Scores.**
(TIFF)

## Author contributions

**Conceptualization:** Diana Akhmedjanova, Tatjana Kanonire.

**Formal analysis:** Andrey Zakharov.

**Investigation:** Diana Akhmedjanova.

**Methodology:** Diana Akhmedjanova, Tatjana Kanonire, Andrey Zakharov.

**Software:** Andrey Zakharov.

**Writing – original draft:** Diana Akhmedjanova, Tatjana Kanonire, Andrey Zakharov.

**Writing – review & editing:** Diana Akhmedjanova, Tatjana Kanonire.

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
