## [Decision Letter · Decision Letter 0]

4 Aug 2025

Moderation analysis of subjective well-being, self-efficacy, and academic performance of 4th grade children in Russia

PLOS ONE

Dear Dr. Akhmedjanova,

Thank you for submitting your manuscript to PLOS ONE. After careful consideration, we feel that it has merit but does not fully meet PLOS ONE’s publication criteria as it currently stands. Therefore, we invite you to submit a revised version of the manuscript that addresses the points raised during the review process.

We look forward to receiving your revised manuscript.

Kind regards,

Musa Adekunle Ayanwale, Ph.D

Academic Editor

PLOS ONE

Journal Requirements:

“This study was conducted with support from the Basic Research Program of HSE University as part of the research project “Longitudinal Study of Factors Related to School Failure.””

5. We note that you have indicated that there are restrictions to data sharing for this study. PLOS only allows data to be available upon request if there are legal or ethical restrictions on sharing data publicly. For more information on unacceptable data access restrictions, please see http://journals.plos.org/plosone/s/data-availability#loc-unacceptable-data-access-restrictions.

Comments from the editorial office:

Upon internal evaluation of the reviews provided, we kindly request you to disregard the reviewer report provided by Reviewer 2. No amendments are required in response to reviewer 2’s comments

Additional Editor Comments:

The authors should streamline the introduction to reduce redundancy, clarify the theoretical contribution of the study, elaborate on the implications of high intercorrelations among constructs, and discuss the practical significance of the observed small effect sizes. Thank you.

Musa Adekunle Ayanwale, Ph.D

Academic Editor

PLOS ONE

Reviewers' comments:

Reviewer's Responses to Questions

**Comments to the Author**

1. Is the manuscript technically sound, and do the data support the conclusions?

Reviewer #1: Yes

Reviewer #2: Yes

Reviewer #3: Yes

2. Has the statistical analysis been performed appropriately and rigorously?

Reviewer #1: Yes

Reviewer #2: Yes

Reviewer #3: Yes

3. Have the authors made all data underlying the findings in their manuscript fully available?

Reviewer #1: Yes

Reviewer #2: Yes

Reviewer #3: Yes

4. Is the manuscript presented in an intelligible fashion and written in standard English?

Reviewer #1: Yes

Reviewer #2: Yes

Reviewer #3: Yes

Reviewer #1: This is a very important paper and definitely has a promising impact in the field and how wellbeing and self-efficacy relates to predict achievement. The author has went to a great length to provide insightful ideas in this paper which makes understanding the concepts easier.

I believe that some clarifications would make the paper more interesting, informative and applicable for other researchers or for practical applications or policy decisions.

General comment

I might be helpful if a direct link to the data is provided. The website has so much information and might not be the best for anyone interested to be looking for a pin in haystack.

I believe the journal follows APA format and the citation and reference list needs to be in the format as well.

If OECD identified five domains, why is the study only using three domains? I wonder if the other two domains are not hedonistic as defined in the paper. Then it was scaled down to two school aspects and affects towards school. The reason for doing this is not very clear.

At the last sentence before self-efficacy title, there is a repetition of study purpose.

This statement is confusing, “the relationship between satisfaction with school and reading (Table 5, Column 2) becomes statistically not different from zero (p<0.05)” and “Finally, in all the models with school fixed effect (Tables 2–5, Column 4), the relationship between both types of wellbeing and both types of students’ outcomes is not statistically significant (p<0.05)”. I believe p < 0.05 means significant. The authors might want to look at the table again.

Major Comment

Did the authors use the validity and reliability in previous studies or the samples in this study?

If the sub-domains of subjective wellbeing and self-efficacy have high correlation like that I wonder if they are truly two factors in the first place. Maybe the authors could provide some factor analysis results to substantiate these two factors.

Stemming from the last comment, running those different analysis could negatively implicate the findings from the study and introduce error.

I wonder if the authors will consider other analytical methods for this study. Latent based methods such as path analysis or others could be useful in this instance.

I wonder how the achievement tests are scored. If it is correct or wrong format, then kr-20 approach should be used instead of alpha and omega methods. I doubt how acceptable the internal consistency of reading test below 0.7 can be said to be acceptable.

I am a bit taken aback when the paper began to mention school fixed effects while the discussion has been about only moderation all along. I wonder if the authors are considering multilevel model initially and something didn’t work along the line, and it was scaled down to moderation analysis.

Reviewer #2: This is a relevant and well-conducted study with a strong sample and useful insights into the role of self-efficacy and well-being in academic performance. However, the manuscript requires major revision. The abstract should clearly report key findings, the literature review should be expanded and structured, the methodology needs clearer justification for certain choices, and other sections that need improvement. With these improvements, the paper will make a valuable contribution to educational research.

Reviewer #3: General Assessment

This manuscript presents as a well-structured study examining the moderating role of domain-specific self-efficacy in the relationship between subjective well-being and academic performance among 4th grade students in Russia. The topic is relevant, particularly given the growing emphasis on holistic educational outcomes. The study is grounded in a solid theoretical framework and contributes to the limited literature on early-grade students in non-Western contexts.

The research questions are clearly articulated and logically derived from the literature.

The use of validated instruments (SSWBS, domain-specific self-efficacy scales, and Progress assessment) is appropriate.

The statistical approach, including OLS regression with interaction terms and school fixed effects, is robust and justified.

The sample size (n = 1,962) is commendable and enhances the generalizability of findings within the studied region.

A few suggestions:

The authors should elaborate on how their findings extend or challenge existing literature, particularly in the Russian context.

Consider discussing the implications of the high correlation between well-being subscales (r = 0.97) and between self-efficacy domains (r = 0.82), which may affect the interpretation of moderation effects.

The introduction could be more concise, with a sharper focus on the research gap and the redundancy between the abstract and introduction could be reduced.

The manuscript is suitable for publication pending minor revisions to enhance clarity and contextual framing. The authors are encouraged to:

Clarify the theoretical contribution of the study.

Expand on the practical significance of small effect sizes.

Streamline the introduction for conciseness.

**Do you want your identity to be public for this peer review?** For information about this choice, including consent withdrawal, please see our Privacy Policy

Reviewer #1: No

Reviewer #2: **Yes: ** Damola Olugbade

Reviewer #3: **Yes: ** Inih Essien

---

## [Author Response · Author response to Decision Letter 1]

30 Sep 2025

Letter of Revisions

This letter of revisions accompanies the revised manuscript Moderation analysis of subjective well-being, self-efficacy, and academic performance of 4th grade children in Russia in response to the Reviewer’s 1 and 3 feedback as well as to editorial comments. We would like to express our deepest gratitude to anonymous reviewers who provided very detailed and constructive feedback on this paper. This letter of revisions provides the detailed response to each comment.

Based on the feedback, we revised the abstract, created separate introduction and literature, and highlighted the research gaps, novelty, and scientific significance of this study in the introduction and discussion sections. All the revisions to the text are represented in the blue font.

In this section, we are responding to each of the comments provided by Reviewer 1:

1. Abstract: The abstract effectively introduces the aim of the study, which is to explore how subjective well-being and domain-specific self-efficacy predict academic achievement in mathematics and reading among elementary school students in Russia. However, the abstract would benefit from being more concise. Specific numerical results (e.g., strength of relationships, effect sizes) are missing, and the conclusion is vague. Including a sentence on practical implications could enhance its scholarly value and clarity. – Thank you for your feedback. We tried to keep the abstract concise, yet added the numerical results and an example of a specific practical implication (p. 2).

2. The keywords: The keywords selected, “subjective well-being,” “self-efficacy,” “mathematics,” “reading,” “elementary school,” and “moderation,” are broadly relevant to the study’s scope. However, the inclusion of terms like “gender differences” and “academic achievement” would strengthen discoverability in academic databases. – Thank you for the suggestion. We added the additional keywords (p. 2).

3. Introduction: The introduction presents a compelling rationale for the research by highlighting the growing emphasis on students’ well-being alongside academic achievement. It provides a well-synthesized background on subjective well-being and self-efficacy, referring to both international and Russian educational contexts. The discussion of prior inconsistent findings helps justify the current research questions. Nevertheless, the introduction would benefit from improved logical flow and more emphasis on the novelty of this study, namely, the focus on elementary students in Russia using moderation analysis. Additionally, some opening paragraphs are slightly philosophical and could be tightened to maintain academic focus. – Thank you for your suggestion. We decided to create introduction as a separate section, where we describe the research gap, novelty, and contributions of this study (p. 3).

4. Literature Review: Although not presented as a distinct section, the literature review is embedded throughout the introduction and early pages. It provides a good overview of key concepts, such as subjective well-being and domain-specific self-efficacy, and discusses findings from major meta-analyses. The review includes important international sources and situates the study within the OECD’s broader educational goals. However, there is a limited critical analysis of Russian-specific literature, and some referenced studies are outdated or not fully detailed (e.g., missing author names). Structuring this into a standalone literature review section with clearer subsections and more recent sources would improve the depth and clarity of the theoretical foundation. - Thank you for your suggestion! We separated the introduction from the literature review and added relevant studies on subjective well-being (p. 5) and self-efficacy (p.8) conducted in Russia. These were the only studies which were conducted in Russia and were done rigorously enough to include them as part of the literature review section. As can be seen, there are a handful studies in general, and only a couple of studies conducted in upper elementary schools. The missing authors’ names in the references was the attempt to blind the manuscript for the review process, since most of them are conducted by the authors of this paper within the last five years. We added all authors’ names at this point (pp. 25 – 27).

5. Methodology: The methodology section is detailed and logically structured. The study design is part of a longitudinal project, although the current data is cross-sectional. Participants were drawn from 40 schools in central Russia, with a sample size of 1,962 fourth-grade students. The use of well-validated instruments to assess well-being and self-efficacy is commendable, and psychometric properties are reported. The statistical approach, including OLS regression and moderation analysis with interaction terms, is appropriate and carefully explained. However, justification for using listwise deletion for missing data instead of more robust methods (like multiple imputation) is lacking. Also, the rationale for not testing both types of self-efficacy in the same model due to high correlation could be better supported with variance inflation factor (VIF) data. Additional information about the sampling process and school selection would strengthen transparency. – Thank you for your suggestion. Information on the sampling procedures and criteria are added on p.9.

We acknowledge the reviewer's concern regarding the use of listwise deletion for handling missing data. Indeed, multiple imputation is considered a robust and preferred method in many contexts. However, our decision to apply listwise deletion was carefully considered based on the characteristics of our data and practical limitations. In our dataset, up to 13% of the values were missing in variables that underpin key constructs such as self-efficacy in math and reading, affect towards school, and satisfaction with school. Importantly, these missing values were due to students skipping specific questions based on which these constructs generated. Unfortunately, the dataset provides only limited auxiliary information beyond these key variables—namely students’ gender, outcome variables (test scores), and location. Multiple imputation requires appropriate auxiliary data that are predictive of both the missingness and the missing values themselves to produce reliable imputations (Graham, 2009). Given the limited auxiliary variables available, we opted for listwise deletion to preserve the validity and interpretability of our findings while acknowledging its limitations. We believe that this approach provides a cautious and justifiable handling of missing data in our study context. We added this information in footnotes on pp. 9-10 to make our decision transparent.

John W. Graham (2009). "Missing Data Analysis: Making It Work in the Real World"._Annual Review of Psychology, 60:549 576._https://www.annualreviews.org/doi/10.1146/annurev.psych.58.110405.085530

In terms of VIF, we included self-efficacy in math as a predictor only in models where math score was the outcome, and self-efficacy in reading only where reading score was the outcome, following the rationale that domain-specific self-efficacy most directly influences outcomes in the corresponding subject. We also chose not to include both satisfaction with school and affect to school in the same model due to their extremely high correlation (r=0,97). When included together, these variables yield a variance inflation factor (VIF) of 19, indicating severe multicollinearity, which compromises the stability and interpretability of regression coefficients.

6. Discussion: The discussion section appropriately interprets the study’s findings and connects them to prior research. It confirms that while subjective well-being is weakly associated with academic outcomes, self-efficacy plays a stronger and more consistent role. The authors provide insightful commentary on gender differences, noting that the moderation effect of self-efficacy is only significant for girls in math. They also suggest that fourth-grade children may not yet have fully developed cognitive abilities to assess their self-efficacy accurately. However, the discussion could be improved by integrating more socio-cultural context into gendered perceptions of math and reading. Additionally, the potential influence of classroom practices and teacher feedback on self-efficacy is mentioned in limitations but not explored deeply in the interpretation of findings. – Thank you for your comment. We added this information in the discussion section on p. 22 and in future research on p. 23.

7. References: The references cited are mostly recent and relevant, including key meta-analyses, foundational theories, and empirical studies on self-efficacy and well-being. However, there are inconsistencies in citation formatting, and some references are incomplete (e.g., anonymous or missing authors). The inclusion of more Russian-language or context-specific studies would enrich the regional relevance of the paper. – Thank you for your observation. We revised and edited the reference list in accordance with the Vancouver citation style.

In this section, we are responding to each of the comments provided by Reviewer #1:

1. General comment: It might be helpful if a direct link to the data is provided. The website has so much information and might not be the best for anyone interested to be looking for a pin in haystack. - Thank you for your observation. Unfortunately, there is no any direct link to the database to provide. Any interested person can find the dataset by entering the following number: 6.0036-2022 or by contacting the first author of this manuscript.

2. I believe the journal follows APA format and the citation and reference list needs to be in the format as well. - Thank you for your observation. However, the journal follows the Vancouver style. Therefore, we could not fix the formatting of citations and references. Please see more information by following this link: https://journals.plos.org/plosone/s/submission-guidelines#loc-reference-style

3. If OECD identified five domains, why is the study only using three domains? I wonder if the other two domains are not hedonistic as defined in the paper. Then it was scaled down to two school aspects and affects towards school. The reason for doing this is not very clear. – Thank you for your comment. We had to select the scales very carefully since this study is part of the longitudinal study and included a huge battery of tests and surveys. As a result, we aimed at keeping the surveys as short as possible but maintaining the rigor. Since these two scales showed high estimates in previous studies and we were interested in well-being in school, we decided to include just satisfaction with school and affect toward school scales.

4. At the last sentence before self-efficacy title, there is a repetition of study purpose. Thank you for your observation. We fixed it.

5. This statement is confusing, “the relationship between satisfaction with school and reading (Table 5, Column 2) becomes statistically not different from zero (p<0.05)” and “Finally, in all the models with school fixed effect (Tables 2–5, Column 4), the relationship between both types of wellbeing and both types of students’ outcomes is not statistically significant (p<0.05)”. I believe p < 0.05 means significant. The authors might want to look at the table again. – Thank you for your comment. We’ve deleted confusing “(p<0.05)” from the text.

6. Major Comment

Did the authors use the validity and reliability in previous studies or the samples in this study? - Thank you for your question. Yes, the validity and reliability studies were conducted previously with all the instruments.

Kanonire T, Federiakin DA, Uglanova IL. Multicomponent framework for students’ subjective well-being in elementary school. Sch Psychol. 2020 Sep;35(5):321-30. doi: 10.1037/spq0000397.

Akhmedjanova D. Domain-specific self-efficacy scales for elementary and middle school students. Psychol Russ-State Art. 2024;17(1):45 66. doi: 10.11621/pir.2024.0103.

7. If the sub-domains of subjective wellbeing and self-efficacy have high correlation like that I wonder if they are truly two factors in the first place. Maybe the authors could provide some factor analysis results to substantiate these two factors. Stemming from the last comment, running those different analysis could negatively implicate the findings from the study and introduce error. I wonder if the authors will consider other analytical methods for this study. Latent based methods such as path analysis or others could be useful in this instance.– Thank you for your comment. Please see the papers cited above for the validity and reliability studies. Also, given the very high correlations between well-being subscales (r = 0.97) and between self-efficacy domains (r = 0.82), we opted not to include both subscales or both domains simultaneously in the same models to avoid issues of multicollinearity. By modeling these constructs separately, we aimed to provide clearer, more reliable estimates of their distinct associations and potential moderation effects. We added this text on p. 12 of the manuscript.

8. I wonder how the achievement tests are scored. If it is correct or wrong format, then kr-20 approach should be used instead of alpha and omega methods. I doubt how acceptable the internal consistency of reading test below 0.7 can be said to be acceptable. – Thank you for your comments, The achievement tests, both for math and reading were presented in the multiple-choice format. We provided some evidence of why the estimates were appropriate for further analyses. Specifically, we added the following text in the methods section (p.10). “Generally, the reliability estimates of > 0.70 are acceptable for conducting further analyses. However, some scholars establish acceptable reliability estimates at 0.65 [35] or at 0.6 [36]. Due to the reading test reliability estimates falling between 0.63 and 0.67, we concluded that the estimates were acceptable to conduct further analyses.”

35. Taber KS. The use of Cronbach’s alpha when developing and reporting research instruments in science education. Res Sci Educ. 2018;48:1273-96. doi: 10.1007/s11165-016-9602-2.

36. Griethuijsen RAL, van Eijck MW, Haste H, den Brok PJ, Skinner NC, Mansour N, et al. Global patterns in students' views of science and interest in science. Res Sci Educ. 2015;45:581-603. doi: 10.1007/s11165-014-9438-6.

9. I am a bit taken aback when the paper began to mention school fixed effects while the discussion has been about only moderation all along. I wonder if the authors are considering multilevel model initially and something didn’t work along the line, and it was scaled down to moderation analysis. – Thank you for your comment. While our article focuses on analyzing moderation effects, we incorporated school fixed effects to some models to control for all observed and unobserved differences between schools, which is a common and rigorous approach in education research. Initially, we also attempted multilevel modeling; however, given the relatively small variance between schools and some challenges with model identification in our sample, this approach proved less suitable. Consequently, we opted for fixed effects, which effectively remove between-school variation.

In this section, we are responding to each of the comments provided by Reviewer #1:

10. The authors should elaborate on how their findings extend or challenge existing literature, particularly in the Russian context. Thank you for your suggestion! We separated the introduction from the literature review and added relevant studies on subjective well-being (p. 5) and self-efficacy (p.8) conducted in Russia. These were the only studies which were conducted in Russia and were done rigorously enough to include them as part of the literature review section. As can be seen, there are a handful studies in general, and only a couple of studies conducted in upper elementary schools. We tried to emphasize this in the introduction on p. 3.

11. Consider discussing the implications of the high correlation

---

## [Decision Letter · Decision Letter 1]

26 Nov 2025

Moderation analysis of subjective well-being, self-efficacy, and academic performance of 4th grade children in Russia

PLOS ONE

Dear Dr. Akhmedjanova,

The introduction should better emphasise the study’s theoretical novelty, while the abstract and discussion should clearly outline the research gap and implications for educational practice. Minor refinements in table presentation and alignment with journal citation formatting are also required.

We look forward to receiving your revised manuscript.

Kind regards,

Musa Adekunle Ayanwale, Ph.D

Academic Editor

PLOS ONE

Journal Requirements:

Reviewers' comments:

Reviewer's Responses to Questions

**Comments to the Author**

Reviewer #1: All comments have been addressed

Reviewer #4: (No Response)

2. Is the manuscript technically sound, and do the data support the conclusions?

Reviewer #1: Yes

Reviewer #4: Yes

3. Has the statistical analysis been performed appropriately and rigorously?

Reviewer #1: Yes

Reviewer #4: Yes

4. Have the authors made all data underlying the findings in their manuscript fully available?

Reviewer #1: Yes

Reviewer #4: Yes

5. Is the manuscript presented in an intelligible fashion and written in standard English?

Reviewer #1: Yes

Reviewer #4: Yes

Reviewer #1: (No Response)

Reviewer #4: 

It is a great pleasure to have reviewed this paper titled “Moderation analysis of subjective well-being, self-efficacy, and academic performance of 4th grade children in Russia”, prepared in the scholarly peer-review style expected for journals such as PLOS ONE or Frontiers in Psychology.

1. Abstract

The abstract effectively summarizes the study’s aims, sample, methods, and key findings. It mentions the statistical approach (moderation analysis via OLS regression) and provides gender-specific insights. The addition of numerical indicators (e.g., SD values) and significance levels strengthens clarity.

Areas for Improvement:

• The abstract should include research gap.

• The abstract could more explicitly highlight the practical implications of the findings (e.g., how teachers or policy makers might use self-efficacy interventions).

• The theoretical contribution—how this study extends existing literature—should be articulated more clearly.

• A statement about the study design type (cross-sectional within a longitudinal project) could improve transparency.

2. Introduction

Strengths:

• Logical progression from global to local context.

• Clear justification of the study’s novelty.

• Appropriate references to foundational theories (Bandura’s self-efficacy, subjective well-being frameworks).

Areas for Improvement:

• The opening paragraphs could be streamlined to avoid philosophical tone and improve focus.

• Greater emphasis on why moderation (not mediation) was chosen theoretically would strengthen conceptual coherence.

• The contribution to the Russian educational policy or classroom practices could be articulated.

3. Literature Review

Strengths:

• Integration of meta-analytic evidence to summarize effect sizes.

• Clarification of construct operationalization (hedonistic approach to well-being).

• Balanced coverage of international and limited Russian studies.

Areas for Improvement:

• The review could include more recent Russian or Eastern European studies to strengthen contextual grounding.

• A conceptual model or diagram could visually depict hypothesized relationships and moderation pathways.

• Greater critical synthesis (rather than descriptive summary) would highlight inconsistencies and justify hypotheses more strongly.

4. Methods

Strengths:

• Appropriate and reliable instruments (SSWBS, domain-specific self-efficacy scales, Progress assessment).

• Reliability coefficients (Cronbach’s α and McDonald’s ω) reported for all measures.

• Clear description of missing-data handling with justification for listwise deletion.

• Transparent explanation of high intercorrelations (r = 0.97; r = 0.82) and exclusion of both subscales simultaneously.

Areas for Improvement:

• Justify the choice of listwise deletion with a brief mention of sensitivity analysis (even a note that results were robust would help).

• Provide sampling frame size and how schools were proportionally selected to improve replicability.

• Mention socio-economic variables or regional educational indicators, if available, as potential covariates.

• Clarify that the reliability values below 0.70 (reading test) are acceptable within exploratory contexts, but indicate implications for interpretation.

5. Data Analysis

Strengths:

• Clear mathematical model presentation.

• Justified exclusion of multicollinear variables.

• Transparent explanation of regression assumptions and potential biases.

Areas for Improvement:

• Consider testing the moderation using interaction plots or simple-slope analyses to visualize effects.

• Include effect-size measures (e.g., ΔR² for moderation).

• Provide rationale for using fixed-effects instead of hierarchical linear modeling (though this is briefly justified, adding intra-class correlation coefficient would reinforce the decision).

6. Results

Strengths:

• Use of standardized coefficients facilitates comparison across models.

• Gender-disaggregated analyses enhance interpretative depth.

• Reporting of p-values, directionality, and magnitudes is consistent.

Areas for Improvement:

• Tables could be condensed (e.g., combine similar models) for better readability.

• Include brief narrative summaries of main findings below tables to reduce cognitive load.

• Highlight practical significance of small effect sizes (β ≈ 0.05–0.10).

7. Discussion

Strengths:

• Thoughtful integration of findings with existing literature.

• Consideration of developmental aspects (e.g., cognitive maturity in self-assessment).

• Recognition of small yet meaningful effects in early education.

Areas for Improvement:

• Expand on socio-cultural explanations for gender differences (e.g., Russian classroom norms, parental expectations).

• Explore implications for teacher practices (feedback, self-efficacy building) more deeply.

• Discuss potential reciprocal causation (academic success enhancing well-being) as a limitation.

Final Recommendation:

Minor revisions are required

Revisions should address:

1. Streamlining the introduction and emphasizing the study’s theoretical novelty.

2. Adding practical implications of findings for educators and curriculum planners.

3. Improving table presentation and discussion clarity.

4. Ensuring citation formatting aligns strictly with journal guidelines.

**Do you want your identity to be public for this peer review?** For information about this choice, including consent withdrawal, please see our Privacy Policy

Reviewer #1: **Yes: ** Daniel O Oyeniran

Reviewer #4: **Yes: ** Oluwaseyi Aina Gbolade Opesemowo

---

## [Author Response · Author response to Decision Letter 2]

29 Dec 2025

Letter of Revisions

This letter of revisions accompanies the revised manuscript Moderation analysis of subjective well-being, self-efficacy, and academic performance of 4th grade children in Russia in response to the Reviewer’s 4 feedback as well as to editorial comments. We would like to express our deepest gratitude to anonymous reviewers who provided very detailed and constructive feedback on this paper. This letter of revisions provides the detailed response to each comment.

Based on the feedback, we revised the abstract, created separate introduction and literature, and highlighted the research gaps, novelty, and scientific significance of this study in the introduction and discussion sections. All the revisions to the text are represented in the blue font.

In this section, we are responding to each of the comments provided by Reviewer 4:

1. Abstract:

• The abstract should include research gap. – Thank you for your feedback. The research gap is added on p.1

• The abstract could more explicitly highlight the practical implications of the findings (e.g., how teachers or policy makers might use self-efficacy interventions). – Thank you. The implications are provided on p. 2

• The theoretical contribution—how this study extends existing literature—should be articulated more clearly. – Thank you. The theoretical contribution is added on p. 2

• A statement about the study design type (cross-sectional within a longitudinal project) could improve transparency. – Thank you. A phrase about the study design was added (p. 1).

2. Introduction:

• The opening paragraphs could be streamlined to avoid philosophical tone and improve focus. – Thank you for your suggestion. We tried to revise accordingly to avoid the philosophical tone. (pp. 3-4)

• Greater emphasis on why moderation (not mediation) was chosen theoretically would strengthen conceptual coherence. – Thank you for your suggestion. We tried to emphasize the moderation effect at the bottom of p.3.

• The contribution to the Russian educational policy or classroom practices could be articulated. – Thank you for your suggestion. We added a sentence about contributions at the end of the introduction section (p. 4).

3. Literature Review:

• The review could include more recent Russian or Eastern European studies to strengthen contextual grounding. - Thank you for your suggestion! Available well-being research from other Eastern European countries mostly focuses on older children (secondary school) (Mihojević et al., 2023; Simic et al., 2024) or even university students (Puiu et al., 2024).These studies are not directly related to the topic of present paper as they consider issues such as inequality (mostly focusing on SES; Mihojević et al., 2023; Simic et al., 2024), physical activity (Janda et al., 2025; Cefai at al., 2025) and others. However, we managed to find several studies looking into self-efficacy, which we added on p.7 (the paragraph in the blue font). We also

• Simić, N., Jović, S., & Petrović, D. (2024). Wellbeing of secondary school students from Serbia: The role of gender, socioeconomic status and ethnic background. In International Psychological Applications Conference and Trends (InPACT 2024), Book of proceedings (pp. 56-60). inScience Press.

• Mihojević, J., Simić, N., & Petrović, D. (2023). The role of gender and socioeconomic status in secondary school academic achievement and wellbeing. Towards a more equitable education: from research to change-Book of Proceedings, 36-43.

• Puiu, S., Udriștioiu, M. T., Petrișor, I., Yılmaz, S. E., Pfefferová, M. S., Raykova, Z., ... & Marekova, E. (2024, July). Students’ well-being and academic engagement: a multivariate analysis of the influencing factors. In Healthcare (Vol. 12, No. 15, p. 1492). MDPI.

• Janda D, Dygrýn J, Chmelík F, Rubín L, Juřicová T, Vorlíček M, Vencálek O, Gába A. Transitions of 24-H Movement Behaviour Profiles From Schooldays to Weekends and Their Associations With Health-Related Quality of Life and Well-Being in Czech Adolescents. Child Care Health Dev. 2025 Jul;51(4):e70121. doi: 10.1111/cch.70121. PMID: 40576241; PMCID: PMC12203759.

• Cefai, C., Barrado, B., Gimenez, G., & Cavioni, V. (2025). Adolescents’ Life Satisfaction, Physical Activity, and the Moderating Role of Gender: A Cross-Country, Multilevel Analysis in 64 Countries. Children, 12(10), 1375. https://doi.org/10.3390/children12101375

• A conceptual model or diagram could visually depict hypothesized relationships and moderation pathways. – Thank you! We added a diagram in Figure 1 on p. 10.

• Greater critical synthesis (rather than descriptive summary) would highlight inconsistencies and justify hypotheses more strongly. - Thank you for your suggestion! We tried to revise the literature review and synthesize the literature. We also added the Current study section (p. 9), where we provide a synthesis of reviewed studies, articulate the research problem, and justify our study.

4. Methodology:

• Justify the choice of listwise deletion with a brief mention of sensitivity analysis (even a note that results were robust would help). Thank you for your comment. We added the following sentence on p. 11: “The listwise deletion was chosen since the sensitivity analyses indicated that the main results were robust to this choice.”

• Provide sampling frame size and how schools were proportionally selected to improve replicability. Thank you for your comment. We added information to the description of the sample on p. 11

• Mention socio-economic variables or regional educational indicators, if available, as potential covariates. – Thank you for this suggestion. Unfortunately, we don’t have those data for this dataset.

• Clarify that the reliability values below 0.70 (reading test) are acceptable within exploratory contexts, but indicate implications for interpretation.– Thank you for this comment. We specified this limitation in the methods section on p. 12. Also, we added a sentence in the limitations section (pp. 22-23) and called for using more robust instruments measuring reading skills in future research.

5. Data analysis:

• Consider testing the moderation using interaction plots or simple-slope analyses to visualize effects. – Thank you for this suggestion. The simple-slope plots were created for the whole sample and included in Appendix A on pp. 30-31.

• Include effect-size measures (e.g., ΔR² for moderation). - – Thank you for this suggestion. The effect size ΔR² for moderation was added in Tables 2-7 and included where appropriate in text of the results sections.

• Provide rationale for using fixed-effects instead of hierarchical linear modeling (though this is briefly justified, adding intra-class correlation coefficient would reinforce the decision). – Thank you for your suggestion. We added the following footnote on p.13 “The intra-class correlation coefficient (ICC), estimated from random-intercept null models, was 0.375 for math scores and 0.115 for reading scores, indicating substantial and moderate between-school variance, respectively. Although this suggests clustering at the school level, particularly for math scores, the primary analyses focused on models with school fixed effects to facilitate interpretation of moderation effects at the individual level. The use of school fixed effects is a common and rigorous approach in education research, as it controls for observed and unobserved differences between schools.”

6. Results:

• Tables could be condensed (e.g., combine similar models) for better readability. – Thank you for this suggestion. We condensed the tables on pp. 15-16.

• Include brief narrative summaries of main findings below tables to reduce cognitive load. - Thank you for this suggestion. We reorganized the tables, so they appear next to the text of the results, describing the estimates from the tables. We hope this has improved the readability of the results.

• Highlight practical significance of small effect sizes (β ≈ 0.05–0.10). – Thank you for this suggestion. We explicitly state the effect sizes are rather small in the discussion section; however, we draw readers’ attention to the role of self-efficacy and how teachers, policy makers, and other stakeholders should invest into developing self-efficacy of elementary school students in implications for practice on p. 24.

7. Discussion:

• Expand on socio-cultural explanations for gender differences (e.g., Russian classroom norms, parental expectations). – Thank you. We expanded on these explanations on p. 22.

• Explore implications for teacher practices (feedback, self-efficacy building) more deeply. – Thank you. We added this information on p. 24.

• Discuss potential reciprocal causation (academic success enhancing well-being) as a limitation. – Thank you for your comment. We added this information in the future research section on p. 23.

Editorial Comments

1. Streamlining the introduction and emphasizing the study’s theoretical novelty. – Thank you. We revised this part on pp. 3- 4.

2. Adding practical implications of findings for educators and curriculum planners. – Thank you. We added this information both in the abstract and implications for practice sections.

3. Improving table presentation and discussion clarity. – We revised the tables and reporting of the results.

4. Ensuring citation formatting aligns strictly with journal guidelines. – Thank you for your observation. We revised and edited the reference list in accordance with the Vancouver citation style.

We would like to thank Reviewer 4 and the editor once again for your time and effort. We really appreciate it.

Sincerely,

Authors

---

## [Editor Report · Decision Letter 2]

6 Jan 2026

Moderation analysis of subjective well-being, self-efficacy, and academic performance of 4th grade children in Russia

PONE-D-25-16123R2

Dear Dr. Akhmedjanova,

We’re pleased to inform you that your manuscript has been judged scientifically suitable for publication and will be formally accepted for publication once it meets all outstanding technical requirements.

Kind regards,

Musa Adekunle Ayanwale, Ph.D

Academic Editor

PLOS One
---

## [Editor Report · Acceptance letter]

PONE-D-25-16123R2

PLOS One

Dear Dr. Akhmedjanova,

I'm pleased to inform you that your manuscript has been deemed suitable for publication in PLOS One. Congratulations! Your manuscript is now being handed over to our production team.

Kind regards,

on behalf of

Dr Musa Adekunle Ayanwale

Academic Editor

PLOS One